# Learning invariant representations and applications to face verification

**Qianli Liao, Joel Z Leibo, and Tomaso Poggio**
Center for Brains, Minds and Machines
McGovern Institute for Brain Research
Massachusetts Institute of Technology
Cambridge MA 02139
lql@mit.edu, jzleibo@mit.edu, tp@ai.mit.edu

## Abstract

One approach to computer object recognition and modeling the brain's ventral stream involves unsupervised learning of representations that are invariant to common transformations. However, applications of these ideas have usually been limited to 2D affine transformations, e.g., translation and scaling, since they are easiest to solve via convolution. In accord with a recent theory of transformation-invariance [1], we propose a model that, while capturing other common convolutional networks as special cases, can also be used with arbitrary identity-preserving transformations. The model's wiring can be learned from videos of transforming objects—or any other grouping of images into sets by their depicted object. Through a series of successively more complex empirical tests, we study the invariance/discriminability properties of this model with respect to different transformations. First, we empirically confirm theoretical predictions (from [1]) for the case of 2D affine transformations. Next, we apply the model to non-affine transformations; as expected, it performs well on face verification tasks requiring invariance to the relatively smooth transformations of 3D rotation-in-depth and changes in illumination direction. Surprisingly, it can also tolerate clutter "transformations" which map an image of a face on one background to an image of the same face on a different background. Motivated by these empirical findings, we tested the same model on face verification benchmark tasks from the computer vision literature: Labeled Faces in the Wild, PubFig [2, 3, 4] and a new dataset we gathered—achieving strong performance in these highly unconstrained cases as well.

## 1   Introduction

In the real world, two images of the same object may only be related by a very complicated and highly nonlinear *transformation*. Far beyond the well-studied 2D affine transformations, objects may rotate in depth, receive illumination from new directions, or become embedded on different backgrounds; they might even break into pieces or deform—melting like Salvador Dali's pocket watch [5]—and still maintain their identity. Two images of the same face could be related by the transformation from frowning to smiling or from youth to old age. This notion of an identity-preserving transformation is considerably more expansive than those normally considered in computer vision. We argue that there is much to be gained from pushing the theory (and practice) of transformation-invariant recognition to accommodate this unconstrained notion of a transformation.

Throughout this paper we use the formalism for describing transformation-invariant hierarchical architectures developed by Poggio et al. (2012). In [1], the authors propose a theory which, they argue, is general enough to explain the strong performance of convolutional architectures across a

wide range of tasks (e.g. [6, 7, 8]) and possibly also the ventral stream. The theory is based on the premise that invariance to identity-preserving transformations is the crux of object recognition.

The present paper has two primary points. First, we provide empirical support for Poggio et al.'s theory of invariance (which we review in section 2) and show how various pooling methods for convolutional networks can all be understood as building invariance since they are all equivalent to special cases of the model we study here. We also measure the model's invariance/discriminability with face-matching tasks. Our use of computer-generated image datasets lets us completely control the transformations appearing in each test, thereby allowing us to measure properties of the representation for each transformation independently. We find that the representation performs well even when it is applied to transformations for which there are no theoretical guarantees—e.g., the clutter "transformation" which maps an image of a face on one background to the same face on a different background.

Motivated by the empirical finding of strong performance with far less constrained transformations than those captured by the theory, in the paper's second half we apply the same approach to face-verification benchmark tasks from the computer vision literature: Labeled Faces in the Wild, Pub-Fig [2, 3, 4], and a new dataset we gathered. All of these datasets consist of photographs taken under natural conditions (gathered from the internet). We find that, despite the use of a very simple classifier—thresholding the angle between face representations—our approach still achieves results that compare favorably with the current state of the art and even exceed it in some cases.

## 2 Template-based invariant encodings for objects unseen during training

We conjecture that achieving invariance to identity-preserving transformations without losing discriminability is the crux of object recognition. In the following we will consider a very expansive notion of 'transformation', but first, in this section we develop the theory for 2D affine transformations[1].

Our aim is to compute a unique *signature* for each image $x$ that is invariant with respect to a group of transformations $G$. We consider the orbit $\{gx \mid g \in G\}$ of $x$ under the action of the group. In this section, $G$ is the 2D affine group so its elements correspond to translations, scalings, and in-plane rotations of the image (notice that we use $g$ to denote both elements of $G$ and their representations, acting on vectors). We regard two images as equivalent if they are part of the same orbit, that is, if they are transformed versions of one another ($x' = gx$ for some $g \in G$).

The orbit of an image is itself invariant with respect to the group. For example, the set of images obtained by rotating $x$ is exactly the same as the set of images obtained by rotating $gx$. The orbit is also unique for each object: the set of images obtained by rotating $x$ only intersects with the set of images obtained by rotating $x'$ when $x' = gx$. Thus, an intuitive method of obtaining an invariant signature for an image, unique to each object, is just to check which orbit it belongs to. We can assume access to a stored set of orbits of template images $\tau_k$; these template orbits could have been acquired by unsupervised learning—possibly by observing objects transform and associating temporally adjacent frames (e.g. [9, 10]).

The key fact enabling this approach to object recognition is this: It is not necessary to have all the template orbits beforehand. Even with a small, sampled, set of template orbits, not including the actual orbit of $x$, we can still compute an invariant signature. Observe that when $g$ is unitary $\langle gx, \tau_k \rangle = \langle x, g^{-1}\tau_k \rangle$. That is, the inner product of the transformed image with a template is the same as the inner product of the image with a transformed template. This is true regardless of whether $x$ is in the orbit of $\tau_k$ or not. In fact, the test image need not resemble any of the templates (see [11, 12, 13, 1]).

Consider $g_t\tau_k$ to be a realization of a random variable. For a set $\{g_t\tau_k, \mid t = 1, ..., T\}$ of images sampled from the orbit of the template $\tau_k$, the distribution of $\langle x, g_t\tau_k \rangle$ is invariant and unique to each object. See [1] for a proof of this fact in the case that $G$ is the group of 2D affine transformations.

Thus, the empirical distribution of the inner products $\langle x, g_t \tau_k \rangle$ is an estimate of an invariant. Following [1], we can use the empirical distribution function (CDF) as the signature:

$$\mu_n^k(x) = \frac{1}{T} \sum_{t=1}^{T} \sigma(\langle x, g_t \tau_k \rangle + n\Delta) \tag{1}$$

where $\sigma$ is a smooth version of the step function ($\sigma(x) = 0$ for $x \leq 0$, $\sigma(x) = 1$ for $x > 0$), $\Delta$ is the resolution (bin-width) parameter and $n = 1, \ldots, N$. Figure 1 shows the results of an experiment demonstrating that the $\mu_n^k(x)$ are invariant to translation and in-plane rotation. Since each face has its own characteristic empirical distribution function, it also shows that these signatures could be used to discriminate between them. Table 1 reports the average Kolmogorov-Smirnov (KS) statistics comparing signatures for images of the same face, and for different faces: $\text{Mean}(KS_{\text{same}}) \sim 0 \implies$ invariance and $\text{Mean}(KS_{\text{different}}) > 0 \implies$ discriminability.

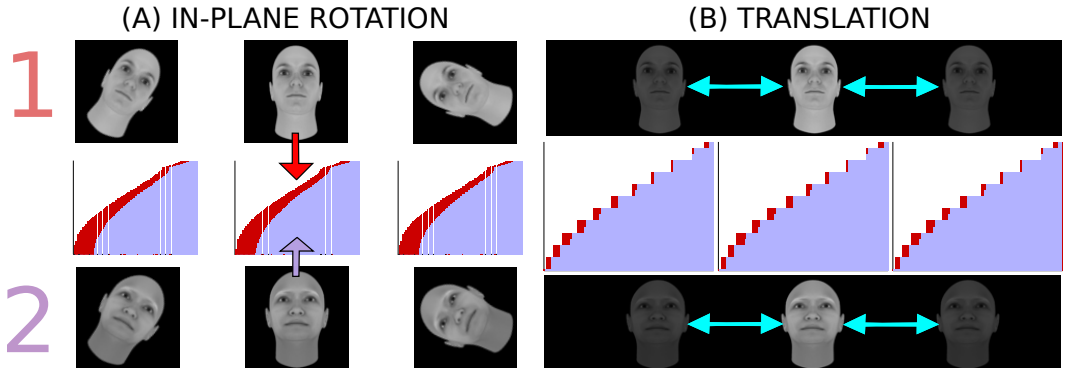

Figure 1: Example signatures (empirical distribution functions—CDFs) of images depicting two different faces under affine transformations. (A) shows in-plane rotations. Signatures for the upper and lower face are shown in red and purple respectively. (B) Shows the analogous experiment with translated faces. Note: In order to highlight the difference between the two distributions, the axes do not start at 0.

Since the distribution of the $\langle x, g_t \tau_k \rangle$ is invariant, we have many choices of possible signatures. Most notably, we can choose any of its statistical moments and these may also be invariant—or nearly so—in order to be discriminative and "invariant for a task" it only need be the case that for each $k$, the distributions of the $\langle x, g_t \tau_k \rangle$ have different moments. It turns out that many different convolutional networks can be understood in this framework[2]. The differences between them correspond to different choices of 1. the set of template orbits (which group), 2. the inner product (more generally, we consider the *template response function* $\Delta_{g\tau_k}(\cdot) := f(\langle \cdot, g_t \tau_k \rangle)$, for a possibly non-linear function $f$—see [1]) and 3. the moment used for the signature. For example, a simple neural-networks-style convolutional net with one convolutional layer and one subsampling layer (no bias term) is obtained by choosing $G$ =translations and $\mu^k(x)$ =mean$(\cdot)$. The $k$-th filter is the template $\tau_k$. The network's nonlinearity could be captured by choosing $\Delta_{g\tau_k}(x) = \tanh(x \cdot g\tau_k)$; note the similarity to Eq. (1). Similar descriptions could be given for modern convolutional nets, e.g. [6, 7, 11]. It is also possible to capture HMAX [14, 15] and related models (e.g. [16]) with this framework. The "simple cells" compute normalized dot products or Gaussian radial basis functions of their inputs with stored templates and "complex cells" compute, for example, $\mu^k(x) = \max(\cdot)$. The templates are normally obtained by translation or scaling of a set of fixed patterns, often Gabor functions at the first layer and patches of natural images in subsequent layers.

## 3 Invariance to non-affine transformations

The theory of [1] only guarantees that this approach will achieve invariance (and discriminability) in the case of affine transformations. However, many researchers have shown good performance of related architectures on object recognition tasks that seem to require invariance to non-affine transformations (e.g. [17, 18, 19]). One possibility is that achieving invariance to affine transformations

is itself a larger-than-expected part of the full object recognition problem. While not dismissing that possibility, we emphasize here that approximate invariance to many non-affine transformations can be achieved as long as the system's operation is restricted to certain *nice* object classes [20, 21, 22].

A nice class with respect to a transformation $G$ (not necessarily a group) is a set of objects that all transform similarly to one another under the action of $G$. For example, the 2D transformation mapping a profile view of one person's face to its frontal view is similar to the analogous transformation of another person's face in this sense. The two transformations will not be exactly the same since any two faces differ in their exact 3D structure, but all faces do approximately share a gross 3D structure, so the transformations of two different faces will not be as different from one another as would, for example, the image transformations evoked by 3D rotation of a chair versus the analogous rotation of a clock. Faces are the prototypical example of a class of objects that is nice with respect to many transformations[3].

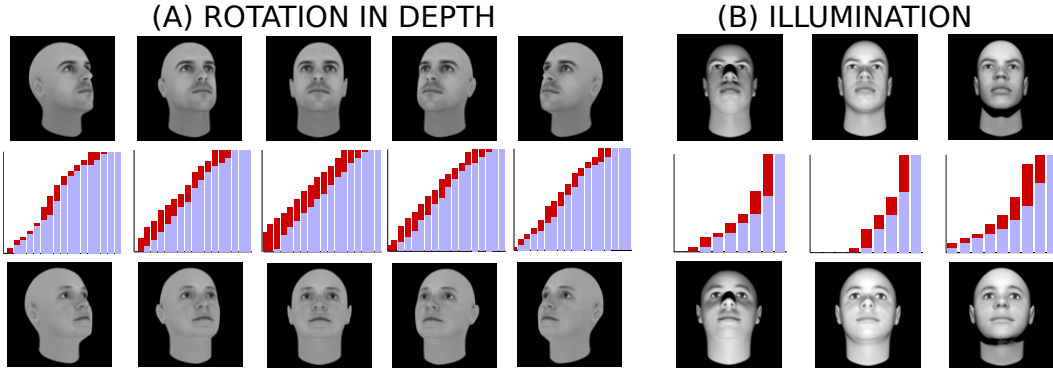

Figure 2: Example signatures (empirical distribution functions) of images depicting two different faces under non-affine transformations: (A) Rotation in depth. (B) Changing the illumination direction (lighting from above or below).

Figure 2 shows that unlike in the affine case, the signature of a test face with respect to template faces at different orientations (3D rotation in depth) or illumination conditions is not perfectly invariant ($KS_{\text{same}} > 0$), though it still tolerates substantial transformations. These signatures are also useful for discriminating faces since the empirical distribution functions are considerably more varied between faces than they are across images of the same face ($\text{Mean}(KS_{\text{different}}) > \text{Mean}(KS_{\text{same}})$, table 1). Table 2 reports the ratios of within-class discriminability (negatively related to invariance) and between-class discriminability for moment-signatures. Lower values indicate both better transformation-tolerance and stronger discriminability.

| Transformation | Mean(KS$_{\text{same}}$) | Mean(KS$_{\text{different}}$) |
|---|---|---|
| Translation | 0.0000 | 1.9420 |
| In-plane rotation | 0.2160 | 19.1897 |
| Out-of-plane rotation | 2.8698 | 5.2950 |
| Illumination | 1.9636 | 2.8809 |

Table 1: Average Kolmogorov-Smirnov statistics comparing the distributions of normalized inner products across transformations and across objects (faces).

| Transformation | MEAN | L1 | L2 | L5 | MAX |
|---|---|---|---|---|---|
| Translation | 0.0000 | 0.0000 | 0.0000 | 0.0000 | 0.0000 |
| In-plane rotation | 0.0031 | 0.0031 | 0.0033 | 0.0042 | 0.0030 |
| Out-of-plane rotation | 0.3045 | 0.3045 | 0.3016 | 0.2923 | 0.1943 |
| Illumination | 0.7197 | 0.7197 | 0.6994 | 0.6405 | 0.2726 |

Table 2: Table of ratios of "within-class discriminability" to "between-class discriminability" for one template $\|\mu(x_i) - \mu(x_j)\|_2$. *within*: $x_i, x_j$ depict the same face, and *between*: $x_i, x_j$ depict different faces. Columns are different statistical moments used for pooling (computing $\mu(x)$).

# 4 Towards the fully unconstrained task

The finding that this templates-and-signatures approach works well even in the difficult cases of 3D-rotation and illumination motivates us to see how far we can push it. We would like to accommodate a totally-unconstrained notion of invariance to identity-preserving transformations. In particular, we investigate the possibility of computing signatures that are invariant to all the task-irrelevant variability in the datasets used for serious computer vision benchmarks. In the present paper we focus on the problem of face-verification (also called *pair-matching*). Given two images of new faces, never encountered during training, the task is to decide if they depict the same person or not.

We used the following procedure to test the templates-and-signatures approach on face verification problems using a variety of different datasets (see fig. 4A). First, all images were preprocessed with low-level features (e.g., histograms of oriented gradients (HOG) [23]), followed by PCA using all the images in the training set and z-score-normalization[4]. At test-time, the $k$-th element of the signature of an image $x$ is obtained by first computing all the $\langle x, g_t \tau_k \rangle$ where $g_t \tau_k$ is the $t$-th image of the $k$-th template person—both encoded by their projection onto the training set's principal components— then pooling the results. We used $\langle \cdot, \cdot \rangle$ = normalized dot product, and $\mu^k(x) = \text{mean}(\cdot)$.

At test time, the classifier receives images of two faces and must classify them as either depicting the same person or not. We used a simple classifier that merely computes the angle between the signatures of the two faces (via a normalized dot product) and responds "same" if it is above a fixed threshold or "different" if below threshold. We chose such a weak classifier since the goal of these simulations was to assess the value of the signature as a feature representation. We expect that the overall performance levels could be improved for most of these tasks by using a more sophisticated classifier[5]. We also note that, after extracting low-level features, the entire system only employs two operations: normalized dot products and pooling.

The images in the Labeled Faces in the Wild (LFW) dataset vary along so many different dimensions that it is difficult to try to give an exhaustive list. It contains natural variability in, at least, pose, lighting, facial expression, and background [2] (example images in fig. 3). We argue here that LFW and the controlled synthetic data problems we studied up to now are different in two primary ways.

First, in unconstrained tasks like LFW, you cannot rely on having seen *all* the transformations of *any* template. Recall, the theory of [1] relies on previous experience with all the transformations of template images in order to recognize test images invariantly to the same transformations. Since LFW is totally unconstrained, any subset of it used for training will never contain all the transformations that will be encountered at test time. Continuing to abuse the notation from section 2, we can say that the LFW database only samples a small subset of $G$, which is now the set of all transformations that occur in LFW. That is, for any two images in LFW, $x$ and $x'$, only a small (relative to $|G|$) subset of their orbits are in LFW. Moreover, $\{g \mid gx \in \text{LFW}\}$ and $\{g' \mid g'x' \in \text{LFW}\}$ almost surely do not overlap with one another[6].

The second important way in which LFW differs from our synthetic image sets is the presence of clutter. Each LFW face appears on many different backgrounds. It is commmon to consider clutter to be a separate problem from that of achieving transformation-invariance, indeed, [1] conjectures that the brain employs separate mechanisms, quite different from templates and pooling—e.g.

attention—toward achieving clutter-tolerance. We set aside those hypotheses for now since the goal of the present work is to explore the limits of the totally unconstrained notion of identity-preserving transformation. Thus, for the purposes of this paper, we consider background-variation as just another transformation. That is, "clutter-transformations" map images of an object on one background to images of the same object on different backgrounds.

We explicitly tested the effects of non-uniform transformation-sampling and background-variation using two new fully-controlled synthetic image sets for face-verification[7]. Figure 3B shows the results of the test of robustness to non-uniform transformation-sampling for 3D rotation-in-depth-invariant face verification. It shows that the method tolerates substantial differences between the transformations used to build the feature representation and the transformations on which the system is tested. We tested two different models of natural non-uniform transformation sampling, in one case (blue curve) we sampled the orbits at a fixed rate when preparing templates, in the other case, we removed connected subsets of each orbit. In both cases the test used the entire orbit and never contained any of the same faces as the training phase. It is arguable which case is a better model of the real situation, but we note that even in the worse case, performance is surprisingly high—even with large percentages of the orbit discarded. Figure 3C shows that signatures produced by pooling over clutter conditions give good performance on a face-verification task with faces embedded on backgrounds. Using templates with the appropriate background size for each test, we show that our models continue to perform well as we increase the size of the background while the performance of standard HOG features declines.

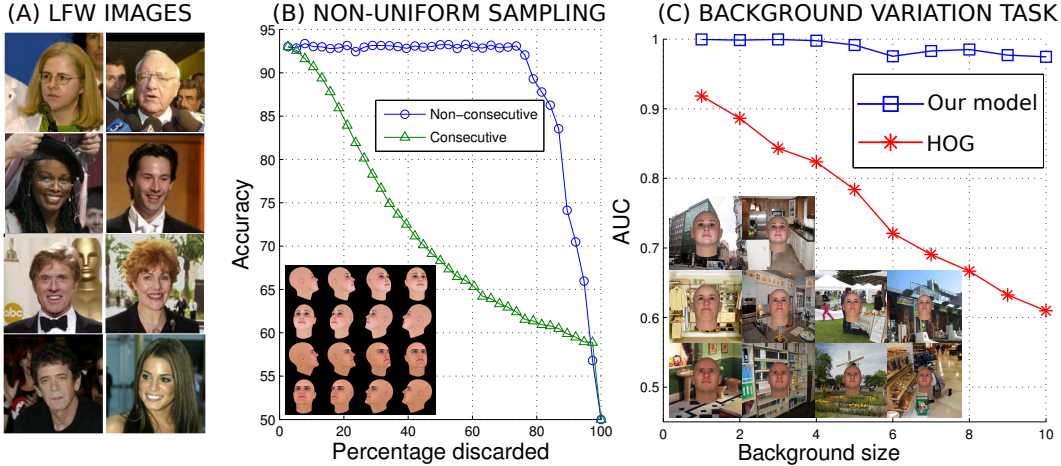

Figure 3: (A) Example images from Labeled Faces in the Wild. (B) Non-uniform sampling simulation. The abscissa is the percentage of frames discarded from each template's transformation sequence, the ordinate is the accuracy on the face verification task. (C) Pooling over variation in the background. The abscissa is the background size (10 scales), and the ordinate is the area under the ROC curve (AUC) for the face verification task.

## 5 Computer vision benchmarks: LFW, PubFig, and SUFR-W

An implication of the argument in sections 2 and 4, is that there needs to be a reasonable number of images sampled from each template's orbit. Despite the fact that we are now considering a totally unconstrained set of transformations, i.e. any number of samples is going to be small relative to $|G|$, we found that approximately 15 images $g_t \tau_k$ per face is enough for all the face verification tasks we considered. 15 is a surprisingly manageable number, however, it is still more images than LFW has for most individuals. We also used the PubFig83 dataset, which has the same problem as LFW, and a subset of the original PubFig dataset. In order to ensure we would have enough images from each template orbit, we gathered a new dataset—SUFR-W[8]—with ∼12,500 images, depicting 450 individuals. The new dataset contains similar variability to LFW and PubFig but tends to have more images per individual than LFW (there are at least 15 images of each individual). The new dataset does not contain any of the same individuals that appear in either LFW or PubFig/PubFig83.

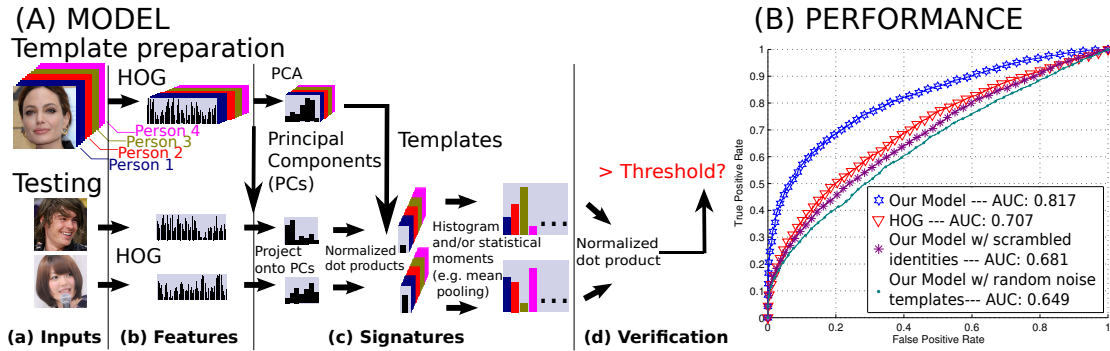

Figure 4: (A) Illustration of the model's processing pipeline. (B) ROC curves for the new dataset using templates from the training set. The second model (red) is a control model that uses HOG features directly. The third (control) model pools over random images in the dataset (as opposed to images depicting the same person). The fourth model pools over random noise images.

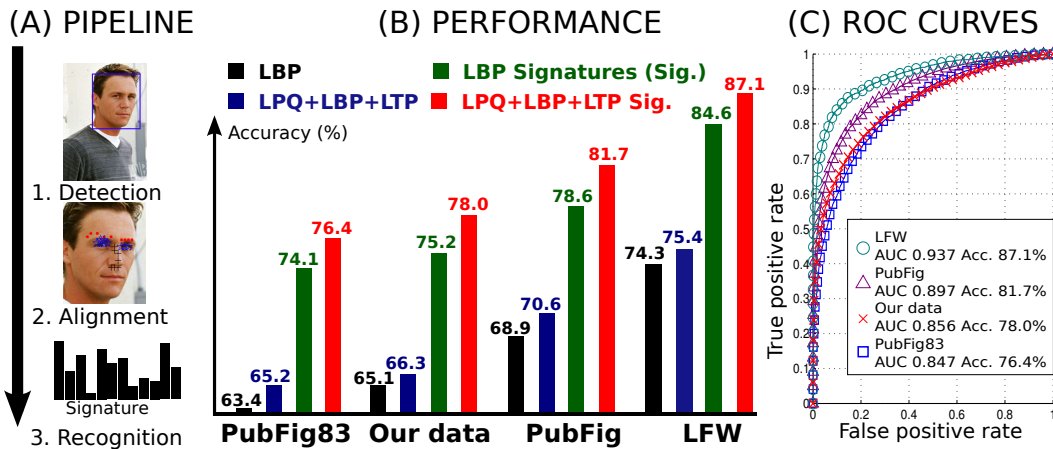

Figure 5: (A) The complete pipeline used for all experiments. (B) The performance of four different models on PubFig83, our new dataset, PubFig and LFW. For these experiments, Local Binary Patterns (LBP), Local Phase Quantization (LPQ), Local Ternary Patterns (LTP) were used [27, 28, 29]; they all perform very similarly to HOG—just slightly better (~1%). These experiments used non-detected and non-aligned face images as inputs—thus the errors include detection and alignment errors (about 1.5% of faces are not detected and 6-7% of the detected faces are significantly mis-aligned). In all cases, templates were obtained from our new dataset (excluding 30 images for a testing set). This sacrifices some performance (~1%) on each dataset but prevents overfitting: *we ran the exact same model on all 4 datasets*. (C) The ROC curves of the best model in each dataset.

Figure 4B shows ROC curves for face verification with the new dataset. The blue curve is our model. The purple and green curves are control experiments that pool over images depicting different individuals, and random noise templates respectively. Both control models performed worse than raw HOG features (red curve).

For all our PubFig, PubFig83 and LFW experiments (Fig. 5), we ignored the provided training data. Instead, we obtained templates from our new dataset. For consistency, we applied the same detection/alignment to all images. The alignment method we used ([30]) produced images that were somewhat more variable than the method used by the authors of the LFW dataset (LFW-a) —the performance of our simple classifier using raw HOG features on LFW is 73.3%, while on LFW-a it is 75.6%.

Even with the very simple classifier, our system's performance still compares favorably with the current state of the art. In the case of LFW, our model's performance exceeds the current state-of-the-art for an unsupervised system (86.2% using LQP — Local Quantized Patterns [31]—Note: these features are not publicly available; otherwise we would have tried using them for preprocess-

ing), though the best supervised systems do better[9]. The strongest result in the literature for face verification with PubFig83[10] is 70.2% [4]—which is 6.2% lower than our best model.

## 6   Discussion

The templates-and-signatures approach to recognition permits many seemingly-different convolutional networks (e.g. ConvNets and HMAX) to be understood in a common framework. We have argued here that the recent strong performance of convolutional networks across a variety of tasks (e.g., [6, 7, 8]) is explained because all these problems share a common computational crux: the need to achieve representations that are invariant to identity-preserving transformations.

We argued that when studying invariance, the appropriate mathematical objects to consider are the orbits of images under the action of a transformation and their associated probability distributions. The probability distributions (and hence the orbits) can be characterized by one-dimensional projections—thus justifying the choice of the empirical distribution function of inner products with template images as a representation for recognition. In this paper, we systematically investigated the properties of this representation for two affine and two non-affine transformations (tables 1 and 2). The same probability distribution could also be characterized by its statistical moments. Interestingly, we found when we considered more difficult tasks in the second half of the paper, representations based on statistical moments tended to outperform the empirical distribution function. There is a sense in which this result is surprising, since the empirical distribution function contains more invariant "information" than the moments—on the other hand, it could also be expected that the moments ought to be less noisy estimates of the underlying distribution. This is an interesting question for further theoretical and experimental work.

Unlike most convolutional networks, our model has essentially no free parameters. In fact, the pipeline we used for most experiments actually has no operations at all besides normalized dot products and pooling (also PCA when preparing templates). These operations are easily implemented by neurons [32]. We could interpret the former as the operation of "simple cells" and the latter as "complex cells"—thus obtaining a similar view of the ventral stream to the one given by [33, 16, 14] (and many others).

Despite the classifier's simplicity, our model's strong performance on face verification benchmark tasks is quite encouraging (Fig. 5). Future work could extend this approach to other objects, and other tasks.

**Acknowledgments**   This material is based upon work supported by the Center for Brains, Minds and Machines (CBMM), funded by NSF STC award CCF-1231216.

## Footnotes

[1]See [1] for a more complete exposition of the theory.

[2]The computation can be made hierarchical by using the signature as the input to a subsequent layer.

[3]It is interesting to consider the possibility that faces co-evolved along with natural visual systems in order to be highly recognizable.

[4]PCA reduces the final algorithm's memory requirements. Additionally, it is much more plausible that the brain could store principal components than directly memorizing frames of past visual experience. A network of neurons with Hebbian synapses (modeled by Oja's rule)—changing its weights online as images are presented—converges to the network that projects new inputs onto the eigenvectors of its past input's covariance [24]. See also [1] for discussion of this point in the context of the templates-and-signatures approach.

[5]Our classifier is unsupervised in the sense that it doesn't have any free parameters to fit on training data. However, our complete system is built using labeled data for the templates, so from that point-of-view it may be considered supervised. On the other hand, we also believe that it could be wired up by an unsupervised process—probably involving the association of temporally-adjacent frames—so there is also a sense in which the entire system could be considered, at least in principle, to be unsupervised. We might say that, insofar as our system models the ventral stream, we intend it as a (strong) claim about what the brain could learn via unsupervised mechanisms.

[6]The brain also has to cope with sampling and its effects can be strikingly counterintuitive. For example, Afraz et al. showed that perceived gender of a face is strongly biased toward male or female at different locations in the visual field; and that the spatial pattern of these biases was distinctive and stable over time for each individual [25]. These perceptual heterogeneity effects could be due to the templates supporting the task differing in the precise positions (transformations) at which they were encountered during development.

[7]We obtained 3D models of faces from FaceGen (Singular Inversions Inc.) and rendered them with Blender (www.blender.org).

[8]See paper [26] for details. Data available at http://cbmm.mit.edu/

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
