[Reviews · NeurIPS 2013]

Submitted by Assigned_Reviewer_4

This paper proposes a new approach for face verification (i.e. for predicting if 2 images represent the same person or not). The approach relies on an auxiliary training set containing several (15 in practice: 15) images of many different persons (in practice: 450). In the following, is it assumed that face images are preprocessed with low level features (HOG) followed by PCA and z-score normalization, giving one template per training image. At test time, the two images to be compared are encoded separately by the following steps: (i) first, the normalized dot product between the template of the test image and each one of the templates of the k-th person is computed (ii) second, the mean value of these dot products gives the k-th component of the representation of the test image (iii) the two previous steps are repeated for each different training persons, resulting in a 450-d representation (as the training set contains 450 persons). The same process is applied independently to the two images to be compared, and the final score is the normalized dot product between their so-computed representations. The experimental validation is done on 4 datasets (PubFig83, a new dataset introduced in the paper, PubFig and LFW). Good results are obtained but no comparisons with recent related work are given. In practice, the results far below from recent competing approaches.

Positive points
===============
- the paper is well written in the sense that, in addition to presenting a new approach for face recognition, it tries to make links with the theory of [1] and to discuss the biological plausibility of the proposed solution
- the paper is clear
- the experiments are interesting and reasonable


Negative points
===============
- the use of auxiliary datasets and more precisely the idea of representing images by the similarities with some sort of prototypes has a long history in computer vision and has received an increasing attention in recent years. E.g. [a] is based on training a separate linear SVM classifier for every exemplar in the training set and representing images by the outputs of these classifiers. More specifically, the use of auxiliary face datasets for transferring some knowledge about face is the central point of several recent papers (e.g. [b-i]). For example, [i] noticed that faces of the same person usually share their top neighbors (in the the auxiliary set). The approach is then to generate a ranking order list by sorting all other faces in the dataset by absolute distance and compute distance of two faces by using their ranking orders. [e] proposed the "associate-predict" model which builds on an extra generic identity
data set, in which each identity contains multiple images
with large intra-personal variation. To compare two faces with different pose/illumination, it first associate one input face with alike identities from the generic identity date set. The associated faces are used to predict the appearance of one input face under the setting of another input face. The proposed approach, even if it also relies on the use of an auxiliary training set, is sufficiently different from the above mentioned approaches, but the fact they are not cited nor discussed raises strong concerns over the paper.
- my second concern is the experimental validation and more precisely the lack of comparisons with previous works. As said before, several competing approaches have been proposed to use auxiliary training sets for face recognition. Some of them learn distance function and are therefore not strictly comparable, but several use the training set to model test images, as the proposed approach does, and are therefore directly comparable. Such comparisons should be given. Moreover, the LFW's results page for the category 'Outside training data in recognition system' (category to which the paper belongs) reports results up to 95% which is much better than the 87.1% obtained by the paper.
- Finally, the experimental validation does not demonstrate if the improvement is due to the method itself (as being an application of the theory of [1]) or is simply due to use of a good auxiliary dataset. A simple set of experiments consisting in representing test images as the normalized dot products with the whole set of test images (images will be represented by a 12,500-dimensional vector, possibly reduced by PCA), without doing any pooling could be useful. If the results are as good as the proposed approach, it would support more the theory of [a] than the theory of [1]. A the end, I have not been able to determine if the proposed face recognition framework is actually an application of [1] or more an exemplar based recognition system [a] (the SVM of [a] being replaced by a simple dot product). Another way to say is to say that there is a big gap between sections 1,2 and 3 where the notion of orbit, geometric transforms, groups, CFD signatures, etc. are at the basis of the recognition process and sections 4 and 5 where there is no more transforms, CDF signatures or whatever but a simple exemplar based recognition approach made by simple dot products.



References
[a] Tomasz Malisiewicz, Abhinav Gupta, Alexei A. Efros. Ensemble of Exemplar-SVMs for Object Detection and Beyond . In ICCV, 2011

[b] Neeraj Kumar, Alexander C. Berg, Peter N. Belhumeur, and Shree K. Nayar. Attribute and Simile Classifiers for Face Verification.
International Conference on Computer Vision (ICCV), 2009.

[c] Vinod Nair and Geoffrey E. Hinton. Rectified Linear Units Improve Restricted Boltzmann Machines. International Conference on Machine Learning (ICML), 2010.

[d] Zhimin Cao, Qi Yin, Xiaoou Tang, and Jian Sun. Face Recognition with Learning-based Descriptor. Computer Vision and Pattern Recognition (CVPR), 2010.

[e] Qi Yin, Xiaoou Tang, and Jian Sun. An Associate-Predict Model for Face Recognition. Computer Vision and Pattern Recognition (CVPR), 2011.

[f] Thomas Berg and Peter N. Belhumeur. Tom-vs-Pete Classifiers and Identity-Preserving Alignment for Face Verification. British Machine Vision Conference (BMVC), 2012.

[g] Dong Chen, Xudong Cao, Liwei Wang, Fang Wen, and Jian Sun. Bayesian Face Revisited: A Joint Formulation. European Conference on Computer Vision (ECCV), 2012.

[h] Dong Chen, Xudong Cao, Fang Wen, and Jian Sun. Blessing of Dimensionality: High-dimensional Feature and Its Efficient Compression for Face Verification. Computer Vision and Pattern Recognition (CVPR), 2013.

[i] Chunhui Zhu, A Rank-Order Distance based Clustering Algorithm for Face Tagging, Fang Wen Jian Sun. Computer Vision and Pattern Recognition (CVPR), 2011.
Summary: An interesting and well written paper on face recognition, weakened by the lack of comparisons/discussions with related recent paper and by an insufficient experimental validation.

Submitted by Assigned_Reviewer_5

The authors empirically show that an existing approach that explains an invariance model for visual recognition system can be extended beyond affine transformation, such as out-of-plane rotation, illumination, and most importantly, altered background clutter. Given that these transformations are major hurdles for building invariant descriptors for objects in imagery, this work is quite interesting since the authors explicitly handled them, one by one, in the framework introduced in [1].
The paper heavily rely on [1], which requires significant elaboration to clearly understand. The paper is very clearly written and quite original and provide good theoretical backgrounds for various aspect for feature design, such as pooling methods.
Summary: In addition to its original contribution of empirically showing the approach in [1] can be extended to various transformation, I think this work will give computer vision researchers, especially who focus on designing/learning representation, a principled guideline regarding to necessary (or sufficient) properties for invariant features.

Submitted by Assigned_Reviewer_6

The paper proposes a very simple and surprisingly effective approach to generating the feature space for the task of face verification (deciding if two face images belong to the same person or not). A set of training templates is chosen ahead of time, and at test time the signature of the image is computed (essentially) by a clever projection onto the training set’s principal components. Additionally, the authors collected a novel face verification dataset with 12,500 images depicting 450 individuals which do not appear in either LFW or PubFig datasets.

POSITIVE:

(1) The proposed method is simple, well-justified and thoroughly analyzed.

(2) The experimental results convincingly demonstrate improvements over state-of-the-art results, as well several baselines, on four datasets (LFW, PubFig and PubFig83, and the newly collected dataset). Notably, the template images were obtained from the new dataset but applied on all the other datasets.

Of particular interest is Figures 3(b), where the accuracy on face verification is evaluated as a function of the number of frames discarded from each template’s transformation sequence, and remains surprisingly even as up to 80% of the (non-consecutive) frames are discarded. Also, Figure 3(c) shows a very impressive invariance to background clutter compared to standard HOG figures.

(3) The paper is exceptionally clear and easy to follow.

COMMENTS:

- Showing ROC curves from 4 different datasets on one set of axes (Figure 5c) is somewhat unconventional
- I look forward to seeing future work on applying this technique to object recognition!
Summary: The paper proposes a new image representation which is invariant to non-affine transformations of the object, and clearly and successfully demonstrates its effectiveness on the task of face verification.
Author Feedback

Author rebuttal: Thanks for taking the time to read our paper. We are quite excited about this project and glad to have received your (mostly) very encouraging assessment. Most of the issues reviewer #4 raised appear to stem from a single misconception which we think will be easy to correct here:

Our approach does not rely on an auxiliary training set.

The logic of the paper is as follows. We begin from the premise that the crux of recognition is the problem of ensuring invariance to identity-preserving transformations without losing discriminability. In section 2, we give the intuition for a theorem from [1] which implies that the distribution of inner products with a stored template is invariant and unique to each object (orbit). This follows from the Cramer-Wold theorem (see [1]). Then we note that the empirical distribution (CDF) of normalized dot products with stored templates is an estimate of that distribution. But there is nothing special about the CDF. As we note in the last paragraph of section 2, any summary statistic (moment) is also an estimate of an invariant. We then explain how convolutional neural networks and HMAX are both special cases of this approach. Or, if you prefer: the theory of [1] explains the strong performance of those networks. In the rest of the paper we continue to apply the same approach to progressively more complicated transformations – moving beyond [1]'s theoretical guarantees. An additional motivation, not stated in the paper, was newer theoretical work which extends [1] and explains many of our empirical findings. Some of the newer theory can be found in more recent updated versions of [1] which are available from the authors' website.

The final section of our paper presents results on real world datasets. The rest of our reply is about this last section since that's where the misconception arose. We think the following numbered points will clear this up.

1. The experiment shown in figure 5 seems to be the cause of the confusion. In that experiment (but not in the other experiments), we trained our model on (the training subset of) a new dataset we gathered which is comparable to Labeled Faces in the Wild (LFW). Let's call it “New Faces in the Wild” (NFW) for the purposes of this review (to preserve anonymity). We only trained the model once. Then we tested it on 4 different datasets: PubFig, PubFig83, LFW, and NFW's validation subset. We emphasize: the exact same model was applied to all 4 datasets. We did not fit any parameters to the specific datasets. This is a much more stringent test than would normally be used. As a result, the accuracy numbers we report are lower than they would be otherwise.

2. We did the “no pooling” control experiment that reviewer 4 suggested. Performance dropped to 74% (with PCA) or 67% (without PCA) on LFW (using templates from NFW). See also the control experiments we reported in figure 4.

3. There is nothing special about our new dataset (NFW). We designed it to be as similar to LFW as possible while being a bit harder than it. The creators of LFW filtered its set of candidate images by the Viola-Jones face detector which, for the most part, only detects frontal or very nearly frontal faces. For NFW, we used the Zhu and Ramanan face detector which works much better with non-frontal faces. Also, NFW is not nearly as well-aligned as LFW(-a). More recently (after our NIPS submission), we tested many different published algorithms on NFW. Many of these algorithms had strong reported LFW performance. Training and testing them on NFW, usually lowered their performance by around 10%. That's not even the most fair comparison to what we did. If these models had to be trained only once and tested on both datasets, as in the test we did for the present paper, then their performance would be even lower than that.

4. We have run the usual LFW test, using LFW training data. We get 86%. We did this first. Then we noticed we were doing much better with PubFig which has a larger number of images per person – but only 60 people not overlapping with LFW. We also noticed that our performance increased as we added individuals to the template set as long as we had at least ~15 images per individual and that PubFig didn't have enough individuals to saturate this effect. NFW has 450 individuals, and at least 15 images of each. So one way to understand our use of NFW is that it was a compromise between LFW, which has many individuals but few images of each, and PubFig, which has many images per individual but too few individuals.

5. Reviewer # 4 wrote: “The proposed approach, even if it also relies on the use of an auxiliary training set, is sufficiently different from the above mentioned approaches, but the fact that they are not cited nor discussed raises strong concerns about the paper”. We emphasize: we do not require outside data. Nevertheless, the papers this reviewer references are interesting and related. We discussed the relationship to that approach more explicitly in an earlier draft, but that section had to be removed in order to comply with the NIPS page limit.

6. Reviewer #4 says that our approach falls into the “outside training data used in recognition system” category for LFW evaluation. The best performance in that category is Chen et al.'s amazing 95%. However, we don't think this is a fair comparison for two reasons. One is the 5th footnote in our paper. The other: hardly any of the systems in that category are as “light” (in all senses) as ours. Our model does nothing besides PCA and dot products. We can perform entire experiments, training once and testing on all 4 datasets, in minutes. Nothing like the hours or days it takes for most of the models in that category.